# The health benefits of volunteering at a free, weekly, 5 km event in the UK: A cross-sectional study of volunteers at *parkrun*

**Steve Haake**[1]*, **Helen Quirk**[2], **Alice Bullas**[1]

**1** The Advanced Wellbeing Research Centre, Sheffield Hallam University, Sheffield, United Kingdom,
**2** School of Health and Related Research, University of Sheffield, Regent Court, Sheffield, United Kingdom

☯ These authors contributed equally to this work.
* s.j.haake@shu.ac.uk

## Abstract

This paper investigates the motives for first participating in *parkrun* and its impact for those who volunteered compared to those who did not volunteer. A cross-sectional survey was emailed to *parkrun* registrants, resulting in 60,680 survey returns from *parkrun* participants who self-identified as volunteers only (n = 681), runners/walkers who volunteered (n = 21,928) or runners/walkers who did not volunteer (38,071). Two survey questions were analysed in this paper: (1) their motives for first participating in *parkrun* as a volunteer or runner/walker; and (2) the perceived impact on their health and wellbeing. More than half of respondents were female and were predominantly from a white ethnic background. Compared to runners/walkers who volunteered, those who volunteered exclusively were older, more likely to be retired and more likely to be inactive at registration. Exclusive volunteers were motivated by wanting to give something back to the community (45.8%), to feel part of a community (26.1%), to help people (24.5%) or because they were unable to run (21.1%). Runners/walkers who volunteered were more likely to volunteer because they felt obliged to (49.3%). A large proportion of exclusive volunteers reported improvements to connections with others such as feeling part of a community (83.5%), the number of new people met (85.2%) and time spent with friends (45.2%). While mental and physical health were ranked low by volunteers as a motive (4.7% and 2.7% respectively), improvements were reported by 54.5 and 29.3% respectively. The data shows that volunteering at *parkrun* without participating as a runner or walker can deliver some of the components of the *Five Ways to Wellbeing* advocated by the NHS. The characteristics of *parkrun* (free, regular, local, accessible and optional) make it a viable social prescribing offer that can be used as a model for other community events seeking to attract volunteers.

## Background

Volunteering is advocated as a way of engaging people in their local communities, improving people's health and wellbeing and building social capital [1]. It has become part of public

**Data Availability Statement:** The dataset (Haake et al. parkrun data Plos Global Public Health 2021. xlsx) is available from the University's public repository (SHURDA) at http://doi.org/10.17032/

shu-180030. Additional data over and above that used in this paper are available through the parkrun Research Board for secondary data analysis through a data sharing agreement: we welcome further analysis of this rich data set.

**Funding:** The parkrun Health and Wellbeing Survey 2018 was partially funded by parkrun to supported the roles of AB and HQ in the delivery of the survey and collection of respondent data. Chrissie Wellington and Mike Graney at parkrun supported the creation of the original survey and facilitated access to parkrun registrant and participation data. For additional research for the manuscript, HQ was funded by the National Institute for Health Research (NIHR) School for Public Health Research (SPHR) post-doctoral launching fellowship, while AB and SH were funded internally by Sheffield Hallam University. The funders had no role in study design, data collection and analysis, decision to publish, or preparation of the manuscript.

**Competing interests:** I have read the journal's policy and the authors of this manuscript have the following competing interests: SH is Chair and AB and HQ members of the parkrun Research Board, an independent body tasked with reviewing research applications to parkrun. All authors are parkrun registrants/participants.

health policy in the UK [2] and across the world [3]. It can take a variety different forms and meanings, but the United Nations (UN) defines volunteering as follows: 1) it is not undertaken for financial reward; 2) it is carried out according to an individual's free-will; and 3) it is of benefit to someone other than the volunteer. It is still recognised, however, that volunteering also brings benefits to the individual [3].

The reported health and wellbeing benefits of volunteering include decreased mortality, improved self-rated health, mental health, life satisfaction, social interaction, healthy behaviours and coping ability [4]. The improvements to wellbeing are said to be particularly evident among older adults [5].

Despite these reported benefits, health improvement is rarely given as a motive for volunteering [4]. People volunteer for different reasons and an understanding of volunteers' motivations is important for their recruitment, management and retention [6, 7]. A common motive tends to be altruistic, or to 'give something back' to a community or organisation [8]. Similarly, social affiliation and solidary motives (connectedness and identification with the group or organisation) are often reported [7]. Motives may vary depending on the activity and whether the volunteering is episodic (short term or task-specific) or sustained [6]. Volunteering also brings benefits to society and the wider group by building trust and reciprocity among community members. The UN has outlined three priority areas to which volunteering contributes: social integration, poverty alleviation and full employment (i.e., enhancing the employability of unemployed people) [3].

The benefits of volunteering at the community level have been promoted through a set of evidence-based public health messages. In 2008, the UK Government's Foresight programme commissioned a study of evidence on cooperative ways to promoting wellbeing in the community. This produced a report by the New Economics Foundations (NEF) called Five Ways to Wellbeing [9] with guidelines which are now promoted by the NHS [10]. The five components are: 1) connect with other people; 2) be physically active; 3) learn new skills; 4) give to others; and 5) be mindful. These guidelines have been applied to volunteering in the community [9] with the key question how to create sustainable and effective schemes that can attract people from diverse communities.

An example of a sustainable and potentially effective scheme that may foster the Five Ways to Wellbeing through volunteering is *parkrun*. *parkrun* is a non-profit organisation that delivers two types of events: 5 km *parkrun*s for those aged 4 and above and 2 km *junior parkruns* for 4- to 14-year-olds and their families. The events are free, weekly and timed and *parkrun* now has over 7 million registrants worldwide. Initially designed as a time trial for runners, it evolved into an event for both runners and walkers, and a place to encourage volunteering. More recently, the World Health Organization, in its Global Action Plan on Physical Activity has called for countries to "implement regular mass-participation initiatives in public spaces, engaging whole communities, to provide free access to enjoyable and affordable, socially and culturally appropriate experiences of physical activity" [11]: *parkrun* was cited as one way of doing this.

*parkruns* are delivered by local teams of volunteers with a network of permanent Ambassadors who help create new events, support outreach programmes, and help oversee activities across 23 countries. Each event has a permanent core team of volunteers who are responsible for the delivery of events supported by a wider network of people who, episodically, carry out event day volunteering duties such as marshalling, timekeeping, scanning barcodes, handing out finish tokens or tail walking [12]. People can volunteer as often or as little as they like and there is no obligation to volunteer [13]. Each volunteering instance is recorded on the *parkrun* database and, like those who run/walk, volunteers are recognised as participants and given t-shirt rewards when they reach certain milestones. Anyone from the age of 4 can volunteer,

with children being supervised by an adult. In the UK, the number of people volunteering at *parkrun* each week is around 20,000 [14], with approximately 175,000 volunteers each year (based on the 3 months prior to October 2021).

The public health impact of *parkrun* on runners has been studied by Stevinson et al. [15] and Grunseit et al. [16], though the impact of *parkrun* on its volunteers is less well understood. Hallett et al. [13] studied runners and walkers at *parkrun* who also volunteered. Hallett et al. found that those volunteered were older, had been registered with *parkrun* for longer and participated more often. The main reasons for not volunteering were *preferring to run*, and *not having got round to it*. To advance our understanding of the health impact of *parkrun*, in 2018, a national survey was carried out in the UK to explore *parkrun's* impact on its participants' health and wellbeing [17].

While most volunteers are runners or walkers, this research paper studies a new volunteer group that has emerged at parkrun: those who volunteer exclusively. It is likely that those who exclusively volunteer are likely to be different to those who also run/walk at *parkrun*, which raises the following research questions: compared to runners/walkers who do or don't volunteer,

1. What is the demographic of those who exclusively volunteer?

2. What are their motives for first volunteering?

3. What is the perceived impact of *parkrun* on their health and wellbeing?

The aim of this study is to investigate those who exclusively volunteer at *parkrun*. The findings will support the retention and growth of this set of volunteers and inform understanding of whether events such as *parkrun* could be a viable social prescribing offer in the promotion of mental wellbeing.

## Materials and methods

The original survey was approved by Sheffield Hallam University Research Ethics Committee (Reference number: ER7034346). Written informed consent was received from all participants via the first page of the online survey. The reporting of this manuscript adheres to established standards for reporting internet-based surveys; The Checklist for Reporting Results of Internet E-Surveys (CHERRIES) [18].

### The survey

A cross-sectional online survey was emailed to 2,318,135 *parkrun* registrants in the UK between 29th October and 3rd December 2018 and collected using Qualtrics online software [19]. The full survey and methods are outlined by Quirk et al. [20]. The web link directed people to an introductory page which contained the participant information sheet and a confirmation box to indicate consent to take part. Only people emailed the web link could access the survey and reminders were sent after one week. There were no incentives offered for taking part in the survey.

Questions were asked in the order presented in the S1 File, with the exception of the International Physical Activity Questionnaire Short Form (IPAQ-SF) [21], which was asked as a final, optional question due to its length and to keep it apart from the other physical activity measures used earlier in the survey. Questions were not randomised, but response choices within some questions were (S1 File).

Certain questions were displayed based on answers to previous questions (adaptive questioning); for example, people who reported being exclusively volunteers did not see questions

about running/walking at *parkrun*. There was a maximum of 47 questions, with an average of 4.3 questions per page and a maximum number of 11 screens (pages) of questions (total question number and page number were shorter due to adaptive questions). The mean completion time was 22 minutes.

Questions were optional (i.e., non-compulsory) except for the question about *parkrun* participation type (to enable the appropriate questions to be presented to the respondent), one question about long-term health conditions and two questions about life satisfaction and happiness. Respondents could go back and forth within the survey to review or change answers. Upon clicking *submit*, answers could not be changed. With consent, partially completed survey responses were saved and data kept for analysis unless the respondent requested removal by contacting the research team.

Responses to the following questions were used in this study:

- ***parkrun* participation type**: participants were asked the following: *Choose one option that best describes your current participation at parkrun*. Options were 1) volunteer only, 2) runner or walker and volunteer (RWV) and 3) runner or walker only (i.e., someone who doesn't volunteer or RWDV). An additional option was *Registered but not yet participated* but were not included in this study.

- **Employment status**: participants were asked: *Which of the following best describes your current employment status*?

- **Ethnicity**: participants were asked: *Which of the following best describes your ethnicity*?

- **Motives for volunteering**: participants self-identified as volunteers or RWV were asked: *What motivated you to first volunteer at* parkrun? [they were asked to select a maximum of three answers out of a possible 26 motives]. Additionally, the choice *Other* allowed a free-text response (not used here). The answer choices were displayed in randomised order to help reduce response bias.

- **Motives for running or walking**: participants self-identified as RWV or RWDV were asked: *What motivated you to first run/walk at* parkrun? [they were asked to select a maximum of three answers out of a possible 20 motives]. Additionally, the choice *Other* allowed a free-text response (not used here). The answer choices were displayed in randomised order to help reduce response bias.

- **Perceived impact of volunteering at *parkrun***: participants self-identified as volunteers or RWV were asked: *Thinking about the impact of parkrun on your health and wellbeing, to what extent has volunteering at parkrun changed*: [list of 17 impacts: much worse/worse/no impact/better/much better]. Additionally, the choice *Other* allowed a free-text response (not used here). The answer choices were displayed in randomised order to help reduce response bias.

- **Perceived impact of running or walking at *parkrun***: participants self-identified as RWV or RWDV were asked: *Thinking about the impact of parkrun on your health and wellbeing, to what extent has running or walking at parkrun changed*: [list of 15 impacts: much worse/worse/no impact/better/much better]. Additionally, the choice *Other* allowed a free-text response (not used here). The answer choices were displayed in randomised order to help reduce response bias.

## Matching data from *parkrun*

During the survey, respondents provided their unique *parkrun* ID number (from their *parkrun* barcode) their date of birth and home *parkrun*. These were matched to their *parkrun*

registration and participation records in the *parkrun* database providing the following additional information for each respondent:

- Date of *parkrun* registration

- Gender provided at *parkrun* registration

- Date of birth provided at *parkrun* registration

- Index of Multiple Deprivation (IMD; derived from postcode given at registration)

- Response to the following questions asked at registration: "Over the last 4 weeks, how often have you done at least 30 minutes of moderate exercise (enough to raise your breathing rate)?" [less than once per week/about once per week/about twice per week/about three times per week/four or more times per week/rather not say/don't know]

- Total number of 5k *parkrun*s completed as a runner/walker and number of volunteer instances.

The number of 5k *parkrun* run/walk or volunteer instances per year were only calculated for those who had been registered for at least a full year.

## Statistics

**Preliminary analysis.** Data was initially validated using Microsoft Excel for Mac (v16.46) using statistical descriptors (counts, mean, median, quartiles, minimum, maximum, skewness and kurtosis). The survey resulted in 100,866 respondents. The following were removed from the analysis: 1) 1,349 respondents who did not consent; 2) 37,039 respondents who consented to view the survey but did not answer any questions; 3) 1,786 respondents who had registered with *parkrun* but not yet participated; and 4) 6 respondents who provided invalid or malicious responses. The dataset used in this paper had 681 who self-identified exclusively as volunteers, 21,928 who identified as runners/walkers who volunteer (RWV) and 38,071 who identified as runners/walkers who don't volunteer (RWDV). This gave a combined data set of 60,680 respondents of which 37.3% were volunteers; 3.0% were volunteers who did not run or walk at parkrun.

Respondents did not answer all questions during the survey or at registration giving different counts for measures reported here; all counts are given. Motives were coded in Microsoft Excel for Mac (v16.49) as *selected* (1) or *not selected* (0) and imported into IBM SPSS Statistics for Mac (v26). Responses to impact measures were coded in a similar manner as *much worse* (1), *worse* (2), *no impact* (3), *better* (4) and *much better* (5).

**Primary analysis.** Statistical analysis was used to compare demographic data between volunteers, RWV and RWDV to allow potential confounding factors to be identified. Where continuous data was non-parametric, medians and the range are given, although means and standard deviations are also given for data that is only marginally so. Continuous data was compared using Mann-Whitney U tests with effect size calculated using $r = Z/\sqrt{n}$, where $Z$ is the standardised test statistic and $n$ the number of ranked respondents [22]. The $\chi^2$ statistic was used to evaluate the differences between the motives for volunteering for volunteers with those for RWV; likewise, the $\chi^2$ statistic was used to evaluate the differences between the impact from volunteering for volunteers with those for RWV and for RWDV. Effect size was estimated using or $\phi_c = \sqrt{(\chi^2/n(k\text{-}1)}$ where $\chi^2$ is the test statistic, $n$ is the number of respondents and $k\text{-}1$ is the number of rows or columns (whichever is the smaller) [23]. Cross tabulation and statistical analysis were carried out using IBM SPSS Statistics for Mac (v26). Statistical significance was set at $p<0.001$ due to the large sample sizes.

## Results

### What are the characteristics of people who volunteer at *parkrun*?

Table 1 shows the characteristics of volunteers (n = 681), RWV (*n* = 21,928) and RWDV (*n* = 38,071). More than half were female (51.1 to 58.1%) and volunteers had a median age of 55.3 years; this dropped by 4.0 years to a median of 51.3 years for RWV and by 8.0 years to 47.3 for RWDV. The maximum age of respondents was over 80 years of age.

About 1 in 10 respondents overall (9.5%) came from the most deprived neighbourhoods (IMD Quartile 1), with 40.1% coming from the least deprived neighbourhoods (IMD Q4). Although there were indications that the proportion in IMD Q1 (most deprived) was higher for volunteers compared to RWV (12.1 v 8.6%), this was non-significant at *p* = 0.021. About 1 in 20 of respondents (5.1%) were inactive at registration (i.e., <1 bout of activity per week) with the proportion for volunteers almost twice that of RWV (8.8% v 4.6%; *p*<0.001) but with a small effect size (0.04). The ethnicity of all respondents was 96.2% white with 3.0% from Black, Asian, or other ethnic backgrounds; there were no differences between sub-groups.

About a third of volunteers were in full-time employment (33.5%) with a further 14.2% in part-time employment; 30.7% were fully retired while only 1.3% were unemployed. RWV had a larger proportion in full-time employment, as did RWDV (51.5 and 56.4% respectively; *p*<0.001) and a smaller proportion who were fully retired (15.2 and 10.5% respectively; *p*<0.001). Effect sizes for these measures were small (less than 0.10).

**How do volunteers engage with parkrun?.** Table 1 shows that volunteers were registered with *parkrun* for a median of 2.36 years, 1.6 years lower than that for RWV (3.93 years; p<0.001, small effect size = 0.08), and 0.6 years higher than that for RWDV (1.74 years; p<0.001, small effect size = 0.03). Volunteers volunteered a median of 11.1 times per year compared to 2.4 times per year for RWV (p<0.001, small to moderate effect size = 0.14) and 0.4 times per year for RWDV despite self-selecting that they didn't volunteer (p<0.001, large effect size = 0.51).

There were 227 volunteers who had been registered for more than a year and had run or walked *parkrun* 2.9 times per year, despite identifying exclusively as a volunteer. This value was less than a sixth of RWV (18.7 per year; *p*<0.001, small to moderate effect size = 0.13). Further analysis (S2 File) showed that during 2018 (i.e., the year of the survey), 48.5% of these volunteers had run/walked no *parkruns* with a further 17.1% doing just one *parkrun* (so that the median for 2018 was one *parkrun*). Some volunteering roles at *parkrun* allow the participant to also complete the run/walk on the same day (e.g., the tail walker volunteer role), which might account for many of these single run/walk instances in 2018. While some volunteers evidently participated at some point as a runner or walker in the past, the participation question asked to *choose one option that best describes your current participation at parkrun*. It is assumed that at the time of the survey, volunteers identified as such.

### What are the motives for first volunteering at *parkrun*?

Table 2 gives the proportions selecting motives for first volunteering at *parkrun* for volunteers and RWV. To allow visual comparison, the proportions for all motives for volunteers compared to RWV are shown in Fig 1A, with the diagonal line indicating where the proportion selecting the motive was equal for each group. The legend relates to the ranking for the full sample shown in Table 2. The data shows that the most selected motive by volunteers was *to give something back to the community*; this was selected by a slightly smaller proportion of volunteers compared to RWV (45.8 v 59.5%; *p*<0.001, small effect size = 0.05). The proportion selecting *as a parkrunner, I felt obliged to volunteer* for RWV was greater than that for

**Table 1. Sample data for participants who were volunteers, runners/walkers who volunteer and runners/walkers.**

| | | Volunteers only | Runners / walkers who volunteer | Runners / walkers who don't volunteer | Total |
|---|---|---|---|---|---|
| Survey responses (n) | | 681 | 21,928 | 38,071 | 60,680 |
| Gender | n | 482 | 17,952 | 28,749 | 47,183 |
| | Female | 58.1% | 52.6% | 51.1% | 51.7% |
| | Male | 41.9% | 47.4% | 48.9% | 48.3% |
| Comparison with volunteers | $\chi^2$ | | 5.65 | 9.36 | |
| | p | | 0.017 | 0.002 | |
| | Effect size $\phi_c$ | | 0.02 | 0.02 | |
| Age (years) | n | 680 | 21,794 | 37,823 | 60,297 |
| | Mean (standard deviation) | 53.8 (14.5) | 50.7 (12.1) | 46.4 (13.4) | 48.0 (13.1) |
| | Median | 55.3 | 51.3 | 47.3 | 49.0 |
| | Range (min–max) | 16.0–84.5 | 16.0–93.5 | 16.0–92.4 | 16.0–93.5 |
| Comparison with volunteers | Z | | 6.55 | 13.5 | |
| | p | | <0.001 | <0.001 | |
| | Effect size r | | 0.04 | 0.07 | |
| Index of multiple deprivation | n | 481 | 17,787 | 28,366 | 46,634 |
| | Quartile 1 | 12.1% | 8.6% | 10.1% | 9.5% |
| | Quartile 2 | 18.1% | 20.0% | 20.6% | 20.3% |
| | Quartile 3 | 32.8% | 30.6% | 29.6% | 30.0% |
| | Quartile 4 | 37.0% | 40.8% | 39.7% | 40.1% |
| Comparison with volunteers | $\chi^2$ | | 9.77 | 5.84 | |
| | p | | 0.021 | 0.120 | |
| | Effect size $\phi_c$ | | 0.02 | 0.01 | |
| Physical activity level at registration | n | 422 | 15,665 | 27,082 | 43,169 |
| | Inactive <1 per week | 8.8% | 4.6% | 5.4% | 5.1% |
| | Active ≈ 1 per week | 13.0% | 10.8% | 11.9% | 11.5% |
| | Active ≈ 2 per week | 23.0% | 22.1% | 23.2% | 22.8% |
| | Active ≈ 3 per week | 29.6% | 34.3% | 33.4% | 33.7% |
| | Active ≥ 4 per week | 25.6% | 28.2% | 26.1% | 26.9% |
| Comparison with volunteers | $\chi^2$ | | 20.97 | 10.96 | |
| | p | | <0.001 | 0.027 | |
| | Effect size $\phi_c$ | | 0.04 | 0.02 | |
| Ethnicity | n | 676 | 21,748 | 37,687 | 60,111 |
| | White | 96.7% | 96.7% | 95.9% | 96.2% |
| | Black, Asian, Other ethnic background | 1.8% | 2.4% | 3.4% | 3.0% |
| | Rather not say | 1.5% | 0.9% | 0.7% | 0.8% |
| Comparison with volunteers | $\chi^2$ | | 3.22 | 11.6 | |
| | p | | 0.200 | 0.003 | |
| | Effect size $\phi_c$ | | 0.01 | 0.02 | |
| Employment status | n | 677 | 21,802 | 37,770 | 60,249 |
| | Full-time paid employment | 33.5% | 51.5% | 56.4% | 54.4% |
| | Part-time paid employment | 14.2% | 14.5% | 13.2% | 13.7% |
| | Fully retired | 30.7% | 15.2% | 10.5% | 12.4% |
| | Self-employed | 8.9% | 9.4% | 9.3% | 9.3% |
| | Student | 3.0% | 1.8% | 3.8% | 3.1% |
| | Unemployed and not working | 1.3% | 1.2% | 1.2% | 1.2% |
| | Other | 8.4% | 6.4% | 5.6% | 5.9% |

*(Continued)*

**Table 1.** (*Continued*)

|  |  | Volunteers only | Runners / walkers who volunteer | Runners / walkers who don't volunteer | Total |
|---|---|---|---|---|---|
| Comparison with volunteers | $\chi^2$ |  | 151.7 | 324.2 |  |
|  | *p* |  | <0.001 | <0.001 |  |
|  | Effect size $\phi_c$ |  | 0.08 | 0.09 |  |
| Years registered with *parkrun* | *n* | 482 | 17,952 | 28,749 | 47,183 |
|  | Median | 2.36 | 3.93 | 1.74 | 2.61 |
|  | Range (min−max) | 0.0–12.7 | 0.0–14.1 | 0.0–13.9 | 0.0–14.1 |
| Comparison with volunteers | *Z* |  | 11.1 | 5.08 |  |
|  | *p* |  | <0.001 | <0.001 |  |
|  | Effect size *r* |  | 0.08 | 0.03 |  |
| *parkruns* volunteered per year | *n* | 353 | 15,908 | 2,568 | 18,829 |
|  | Median | 11.08 | 2.36 | 0.41 | 1.92 |
|  | Range (min−max) | 0.2–76.5 | 0.1–80.3 | 0.1–29.4 | 0.1–80.3 |
| Comparison with volunteers | *Z* |  | 17.6 | 27.6 | 27.6 |
|  | *p* |  | <0.001 | <0.001 | <0.001 |
|  | Effect size *r* |  | 0.14 | 0.51 | 0.51 |
| *parkruns* run/walked per year | *n* | 227 | 16,220 | 17,991 | 34,438 |
|  | Median | 2.88 | 18.71 | 6.06 | 11.22 |
|  | Range (min−max) | 0.1–50.2 | 0.1–55.6 | 0.1–55.6 | 0.1–55.6 |
| Comparison with volunteers | *Z* |  | 17.2 | 6.57 |  |
|  | *p* |  | <0.001 | <0.001 |  |
|  | Effect size *r* |  | 0.13 | 0.05 |  |

Comparisons: categorical data, Chi-squared test; continuous data, Mann-Witney U test.

volunteers (49.3 v 7.5%; p<0.001, small to moderate effect size = 0.14). Similarly, the proportion selecting *to fulfil a moral duty* for RWV was greater than that for volunteers (16.9 v 3.5%; *p*<0.001, small effect size = 0.06) and for *wanted a rest/recovery day* (10.8 v 1.4%; *p*<0.001, small effect size = 0.05).

Around a quarter of all volunteers selected *to help people* and *to feel part of a community* with no significant difference between volunteers and RWV (*p* = 0.070 and 0.300 respectively). Where the number of respondents was >50, volunteers were more likely than RWV to select the following (*p*<0.001): *unable to walk or run* (21.1 v 14.0%, small effect size = 0.04), *it was a good use of my time* (15.7 v 9.5%, small effect size = 0.04), *to meet new people* (13.4 v 6.2%, small effect size = 0.05), *to spend time outdoors* (17.2 v 4.4, small to moderate effect size = 0.10), *my friends family or colleagues encouraged me to* (10.7 v 2.94%, small effect size = 0.08) and *to spend time with family* (17.5 v 2.74%, small to moderate effect size = 0.15).

Few chose *to contribute to my fitness* or *to improve my physical health* and even fewer selected motives relating to new skills (*to develop my skills*, *it was part of a volunteering programme or course (e.g. Duke of Edinburgh)*, *to improve my CV/employability*). Very few were motivated *to improve or manage my health condition* or because *a health professional advised me to*.

Table 3 compares the motives for first volunteering by volunteers with those for running/walking by RWDV. The list contains the 13 measures common to each question (S1 File) with the results shown diagrammatically in Fig 1B, using the same legend as Table 2 and Fig 1A. Compared to RWDV, volunteers were more motivated by *feeling part of a community* (26.0 v

**Table 2. Motives for first volunteering at *parkrun* for those who exclusively volunteer and runners/walkers who volunteer.**

| | (Values in *italics* n<50) | Volunteers only | Runners / walkers who volunteer | All | $\chi^2$ | p | Effect size |
|---|---|---|---|---|---|---|---|
| | n | 664 | 21,416 | 22,080 | | | |
| 1 | to give something back to the community | 45.8% | 59.5% | 59.1% | 50.1 | <0.001 | 0.05 |
| 2 | as a *parkrunner*, I felt obliged to volunteer | 7.5% | 49.3% | 48.1% | 450.5 | <0.001 | 0.14 |
| 3 | to help people | 24.5% | 27.7% | 27.6% | 3.29 | 0.070 | 0.01 |
| 4 | to feel part of a community | 26.1% | 24.3% | 24.4% | 1.08 | 0.300 | 0.01 |
| 5 | to fulfil a moral duty | *3.5%* | 16.9% | 16.5% | 84.8 | <0.001 | 0.06 |
| 6 | unable to walk or run (e.g., due to injury illness or health condition) | 21.1% | 14.0% | 14.2% | 26.4 | <0.001 | 0.04 |
| 7 | wanted a rest / recovery day | *1.4%* | 10.8% | 10.5% | 61.1 | <0.001 | 0.05 |
| 8 | it was a good use of my time | 15.7% | 9.3% | 9.5% | 30.1 | <0.001 | 0.04 |
| 9 | to work with a team of people | 10.8% | 8.7% | 8.8% | 3.74 | 0.053 | 0.01 |
| 10 | to meet new people | 13.4% | 6.2% | 6.5% | 54.6 | <0.001 | 0.05 |
| 11 | to gain a sense of personal achievement | 5.3% | 5.4% | 5.4% | 0.01 | 0.917 | 0.00 |
| 12 | to spend time outdoors | 17.2% | 4.4% | 4.8% | 232.9 | <0.001 | 0.10 |
| 13 | to spend time with friends | *7.4%* | 3.5% | 3.7% | 26.9 | <0.001 | 0.04 |
| 14 | my friends, family or colleagues encouraged me to | 10.7% | 2.9% | 3.2% | 127.1 | <0.001 | 0.08 |
| 15 | to spend time with family | 17.5% | 2.7% | 3.1% | 464.4 | <0.001 | 0.15 |
| 16 | to improve my happiness | *4.2%* | 2.4% | 2.5% | 8.88 | 0.003 | 0.02 |
| 17 | to improve my confidence | *2.7%* | 2.0% | 2.0% | 1.58 | 0.209 | 0.01 |
| 18 | to improve my mental health | *4.7%* | 1.6% | 1.6% | 38.7 | <0.001 | 0.04 |
| 19 | to develop my skills | *1.2%* | 1.1% | 1.1% | 0.1 | 0.757 | 0.00 |
| 20 | to contribute to my fitness | *2.7%* | 0.9% | 1.0% | 22.5 | <0.001 | 0.03 |
| 21 | to improve my physical health | *2.9%* | 0.5% | 0.6% | 57.7 | <0.001 | 0.05 |
| 22 | to gain recognition for my accomplishments | *0.2%* | 0.4% | 0.4% | 0.93 | 0.335 | 0.00 |
| 23 | to improve or manage my health condition, disability or illness | *2.4%* | 0.3% | 0.4% | 80.8 | <0.001 | 0.06 |
| 24 | it was part of a volunteering programme or course (e.g. Duke of Edinburgh) | *2.3%* | 0.3% | 0.4% | 58.7 | <0.001 | 0.05 |
| 25 | to improve my CV / employability | *1.2%* | 0.3% | 0.3% | 15.9 | <0.001 | 0.03 |
| 26 | a health professional advised me to | *0.2%* | *0.0%* | 0.0% | 2.03 | 0.150 | 0.01 |

8.0%, small to moderate effect size = 0.10), *spending time with family* (17.4 v 7.7%, small effect size = 0.05), *spending time outdoors* (17.1 v 10.2%, small effect size = 0.03) and *to meet new people* (13.4 v 2.9%, small effect size = 0.08). Conversely, volunteers were less likely than RWDV to be motivated by *fitness* (2.7 v 55.9%, small to moderate effect size = 0.14), *physical health* (2.9 v 38.0%, small to moderate effect size = 0.10) and a *sense of personal achievement* (5.3 v 27.6%, small effect size = 0.07).

## What are the perceived impacts of volunteering at *parkrun*?

Table 4 shows the proportions of volunteers and RWV reporting *much worse* to *much better* for the 17 volunteering impact measures; the data is shown diagrammatically in Fig 2. Respondents tended to report *no impact*, *better* or *much better* with less than 1% reporting *worse* or *much worse* (the exception being *the amount of time you spend with family* at 6.5 and 0.3% respectively). A majority of categories for *worse* and *much worse* had counts less than 5 and were combined with *no impact* for $\chi^2$ tests.

The proportions selecting *better + much better* were largest in the full sample for *how much you feel part of a community* (84.3%), *the number of new people you meet* (79.2%), *your sense of*

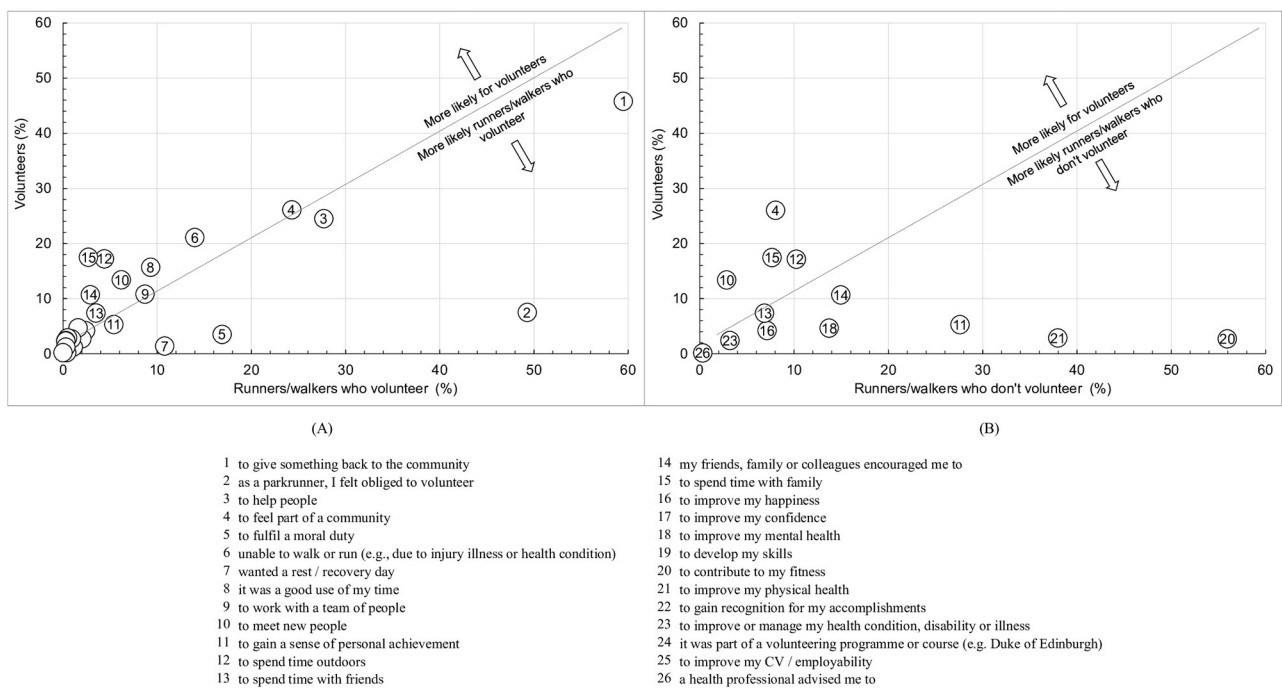

**Fig 1. Motives for first participating in *parkrun* for volunteers who exclusively volunteer compared to: (A) runners/walkers who volunteer; (B) runners/walkers who don't volunteer.** See legend for specific motives.

*personal achievement* (72.5%), *your ability to help people* (72.3%) and *happiness* (68.7%). Fig 3 compares the impact reported by volunteers with RWV for the impact categories *much worse + worse + no impact*, *better*, *much better* and *better + much better*. The diagonal line indicates where the proportions for volunteers and RWV were equal. Linear regression lines are also shown, and all have *p*<0.001. Shaded points indicate those measures where the distribution for volunteers were significantly different to that for RWV.

**Table 3. Motives for first volunteering at *parkrun* for those who exclusively volunteer and for first participating as a runner/walker for those who don't volunteer.**

|  | (Values in *italics* n<50) | Volunteers only | Runners / walkers who don't volunteer | All | $\chi^2$ | *p* | $\phi$ |
|---|---|---|---|---|---|---|---|
|  | *n* | 664 | 37,668 | 38,332 |  |  |  |
| 20 | to contribute to my fitness | *2.7%* | 55.9% | 55.0% | 747.2 | <0.001 | 0.14 |
| 21 | to improve my physical health | *2.9%* | 38.0% | 37.4% | 343.9 | <0.001 | 0.10 |
| 11 | to gain a sense of personal achievement | *5.3%* | 27.6% | 27.2% | 164.3 | <0.001 | 0.07 |
| 14 | my friends, family or colleagues encouraged me to | 10.7% | 15.0% | 14.9% | 9.4 | 0.002 | 0.02 |
| 18 | to improve my mental health | *4.7%* | 13.7% | 13.6% | 100.1 | <0.001 | 0.05 |
| 12 | to spend time outdoors | 17.1% | 10.2% | 10.4% | 33.7 | <0.001 | 0.03 |
| 4 | to feel part of a community | 26.0% | 8.0% | 8.4% | 395.8 | <0.001 | 0.10 |
| 15 | to spend time with family | 17.4% | 7.7% | 7.8% | 87.1 | <0.001 | 0.05 |
| 16 | to improve my happiness | *4.2%* | 7.1% | 7.1% | 8.5 | 0.004 | 0.02 |
| 13 | to spend time with friends | *7.4%* | 6.9% | 6.9% | 0.3 | 0.609 | 0.00 |
| 23 | to improve or manage my health condition, disability or illness | *2.4%* | 3.2% | 3.2% | 1.3 | 0.249 | 0.01 |
| 10 | to meet new people | 13.4% | 2.9% | 3.0% | 246.6 | <0.001 | 0.08 |
| 26 | a health professional advised me to | *0.2%* | 0.3% | 0.3% | 0.6 | 0.450 | 0.00 |

**Table 4. Perceived impact of volunteering at *parkrun* using the question "*Thinking about the impact of parkrun on your health and wellbeing, to what extent has volunteering at parkrun changed [your]*".**

| | Measure | *n* | Worse + much worse | Much worse | Worse | No impact | Better | Much better | Better + Much Better | $\chi^2$ | *p* | $\phi_c$ |
|---|---|---|---|---|---|---|---|---|---|---|---|---|
| | | | | | | | | | **Volunteers** | | $\chi^2$ tests | |
| 1 | How much you feel part of a community | 632 | 16.5% | 0.2% | 0.3% | 16.0% | 61.6% | 22.0% | 83.5% | | | |
| 2 | Number of new people you meet | 630 | 14.8% | 0.2% | 0.3% | 14.3% | 60.6% | 24.6% | 85.2% | | | |
| 3 | Sense of personal achievement | 630 | 25.1% | 0.0% | 0.2% | 24.9% | 58.9% | 16.0% | 74.9% | | | |
| 4 | Ability to help people | 628 | 31.2% | 0.0% | 0.0% | 31.2% | 54.0% | 14.8% | 68.8% | | | |
| 5 | Happiness | 626 | 28.8% | 0.2% | 0.8% | 27.8% | 57.7% | 13.6% | 71.2% | | | |
| 6 | Ability to fulfil moral duties | 621 | 50.7% | 0.3% | 0.3% | 50.1% | 40.4% | 8.9% | 49.3% | | | |
| 7 | Amount of time you spend outdoors | 628 | 26.8% | 0.0% | 0.3% | 26.4% | 60.5% | 12.7% | 73.2% | | | |
| 8 | Mental health | 626 | 45.5% | 0.3% | 1.3% | 43.9% | 43.8% | 10.7% | 54.5% | | | |
| 9 | Ability to work with a team | 629 | 48.2% | 0.0% | 0.3% | 47.9% | 40.2% | 11.6% | 51.8% | | | |
| 10 | Confidence | 627 | 52.8% | 0.2% | 0.6% | 52.0% | 36.2% | 11.0% | 47.2% | | | |
| 11 | Amount of time you spend with friends | 628 | 54.8% | 0.0% | 0.6% | 54.1% | 37.9% | 7.3% | 45.2% | | | |
| 12 | Ability to gain recognitions for your accomplishments | 624 | 61.7% | 0.2% | 0.6% | 60.9% | 30.8% | 7.5% | 38.3% | | | |
| 13 | Skills | 620 | 61.1% | 0.2% | 0.2% | 60.8% | 32.4% | 6.5% | 38.9% | | | |
| 14 | Physical health | 628 | 64.6% | 0.0% | 1.1% | 63.5% | 29.3% | 6.1% | 35.4% | | | |
| 15 | Fitness | 624 | 70.7% | 0.2% | 1.0% | 69.6% | 24.7% | 4.6% | 29.3% | | | |
| 16 | Amount of time you spend with family | 626 | 67.1% | 0.0% | 5.1% | 62.0% | 24.6% | 8.3% | 32.9% | | | |
| 17 | CV/employability | 606 | 83.2% | 0.5% | 0.5% | 82.2% | 13.7% | 3.1% | 16.8% | | | |
| | *Runners/walkers who volunteer* | | | | | | | | | | | |
| 1 | How much you feel part of a community | 20,245 | 15.6% | 0.0% | 0.1% | 15.5% | 65.1% | 19.2% | 84.4% | | | |
| 2 | Number of new people you meet | 20,257 | 21.0% | 0.0% | 0.0% | 21.0% | 64.0% | 15.0% | 79.0% | | | |
| 3 | Sense of personal achievement | 20,244 | 27.6% | 0.0% | 0.1% | 27.4% | 59.4% | 13.0% | 72.4% | | | |
| 4 | Ability to help people | 20,206 | 27.6% | 0.0% | 0.0% | 27.6% | 59.5% | 12.9% | 72.4% | | | |
| 5 | Happiness | 20,216 | 31.3% | 0.0% | 0.2% | 31.1% | 58.9% | 9.7% | 68.7% | | | |
| 6 | Ability to fulfil moral duties | 20,235 | 33.9% | 0.0% | 0.1% | 33.8% | 55.2% | 10.9% | 66.1% | | | |
| 7 | Amount of time you spend outdoors | 20,211 | 38.5% | 0.0% | 0.1% | 38.4% | 52.2% | 9.3% | 61.5% | | | |
| 8 | Mental health | 20,240 | 49.0% | 0.0% | 0.2% | 48.7% | 43.0% | 8.0% | 51.0% | | | |
| 9 | Ability to work with a team | 20,214 | 49.1% | 0.0% | 0.1% | 49.0% | 42.4% | 8.6% | 50.9% | | | |
| 10 | Confidence | 20,207 | 50.5% | 0.0% | 0.2% | 50.2% | 41.4% | 8.1% | 49.5% | | | |
| 11 | Amount of time you spend with friends | 20,214 | 61.1% | 0.0% | 1.0% | 60.1% | 32.7% | 6.2% | 38.9% | | | |
| 12 | Ability to gain recognitions for your accomplishments | 20,203 | 62.1% | 0.0% | 0.2% | 61.9% | 32.3% | 5.6% | 37.9% | | | |
| 13 | Skills | 20,206 | 63.0% | 0.0% | 0.1% | 62.9% | 32.1% | 4.9% | 37.0% | | | |
| 14 | Physical health | 20,212 | 74.0% | 0.0% | 0.5% | 73.6% | 20.5% | 5.5% | 26.0% | | | |
| 15 | Fitness | 20,206 | 77.5% | 0.0% | 0.9% | 76.5% | 16.9% | 5.6% | 22.5% | | | |
| 16 | Amount of time you spend with family | 20,217 | 81.3% | 0.3% | 6.6% | 74.4% | 15.6% | 3.2% | 18.7% | | | |
| 17 | CV/employability | 20,035 | 87.6% | 0.1% | 0.1% | 87.4% | 10.2% | 2.2% | 12.4% | | | |
| | *All who volunteer* | | | | | | | | | | | |
| 1 | How much you feel part of a community | 20,889 | 0.2% | 0.0% | 0.0% | 0.2% | 0.7% | 0.2% | 84.3% | 3.9 | 0.143 | 0.01 |
| 2 | Number of new people you meet | 20,875 | 0.2% | 0.0% | 0.0% | 0.2% | 0.6% | 0.2% | 79.2% | 49.4 | <0.001 | 0.05 |
| 3 | Sense of personal achievement | 20,874 | 0.3% | 0.0% | 0.0% | 0.3% | 0.6% | 0.1% | 72.5% | 5.6 | 0.061 | 0.02 |
| 4 | Ability to help people | 20,863 | 0.3% | 0.0% | 0.0% | 0.3% | 0.6% | 0.1% | 72.3% | 7.8 | 0.020 | 0.02 |
| 5 | Happiness | 20,842 | 0.3% | 0.0% | 0.0% | 0.3% | 0.6% | 0.1% | 68.7% | 10.5 | 0.005 | 0.02 |
| 6 | Ability to fulfil moral duties | 20,835 | 0.3% | 0.0% | 0.0% | 0.3% | 0.5% | 0.1% | 65.6% | 76.0 | <0.001 | 0.06 |
| 7 | Amount of time you spend outdoors | 20,858 | 0.4% | 0.0% | 0.0% | 0.4% | 0.5% | 0.1% | 61.8% | 37.9 | <0.001 | 0.04 |

*(Continued)*

**Table 4.** (Continued)

| | Measure | *n* | Worse + much worse | Much worse | Worse | No impact | Better | Much better | Better + Much Better | Volunteers χ² | | |
|---|---|---|---|---|---|---|---|---|---|---|---|---|
| | | | | | | | | | | χ² | *p* | $\phi_c$ |
| 8 | Mental health | 20,837 | 0.5% | 0.0% | 0.0% | 0.5% | 0.4% | 0.1% | 51.1% | 7.2 | 0.027 | 0.02 |
| 9 | Ability to work with a team | 20,869 | 0.5% | 0.0% | 0.0% | 0.5% | 0.4% | 0.1% | 51.0% | 7.2 | 0.027 | 0.02 |
| 10 | Confidence | 20,834 | 0.5% | 0.0% | 0.0% | 0.5% | 0.4% | 0.1% | 49.4% | 10.8 | 0.004 | 0.02 |
| 11 | Amount of time you spend with friends | 20,842 | 0.6% | 0.0% | 0.0% | 0.6% | 0.3% | 0.1% | 39.1% | 10.3 | 0.006 | 0.02 |
| 12 | Ability to gain recognitions for your accomplishments | 20,830 | 0.6% | 0.0% | 0.0% | 0.6% | 0.3% | 0.1% | 37.9% | 4.6 | 0.100 | 0.02 |
| 13 | Skills | 20,823 | 0.6% | 0.0% | 0.0% | 0.6% | 0.3% | 0.0% | 37.1% | 3.1 | 0.215 | 0.01 |
| 14 | Physical health | 20,840 | 0.7% | 0.0% | 0.0% | 0.7% | 0.2% | 0.1% | 26.3% | 30.5 | <0.001 | 0.04 |
| 15 | Fitness | 20,841 | 0.8% | 0.0% | 0.0% | 0.8% | 0.2% | 0.1% | 22.7% | 25.6 | <0.001 | 0.04 |
| 16 | Amount of time you spend with family | 20,832 | 0.8% | 0.0% | 0.1% | 0.7% | 0.2% | 0.0% | 19.2% | 94.9 | <0.001 | 0.07 |
| 17 | CV/employability | 20,641 | 0.9% | 0.0% | 0.0% | 0.9% | 0.1% | 0.0% | 12.5% | 10.7 | 0.005 | 0.02 |

Measures are ranked largest to smallest by *better + much better* for all volunteers and χ² tests used a combined category for *no impact*, *worse* and *much worse* due to counts less than five in the latter two categories. Values shown for volunteers, runners/walkers who volunteer and all who volunteer.

Volunteers were more likely than RWV to report *better* for measures with low scores (Fig 3B) but were more likely than RWV to report *much better* for almost all measures (Fig 3C), with a gradient of 1.19 (*p*<0.001). Overall, volunteers were more likely than RWV to report *better + much better* from volunteering (Fig 3D) for measures such as *physical health* (35.4 v

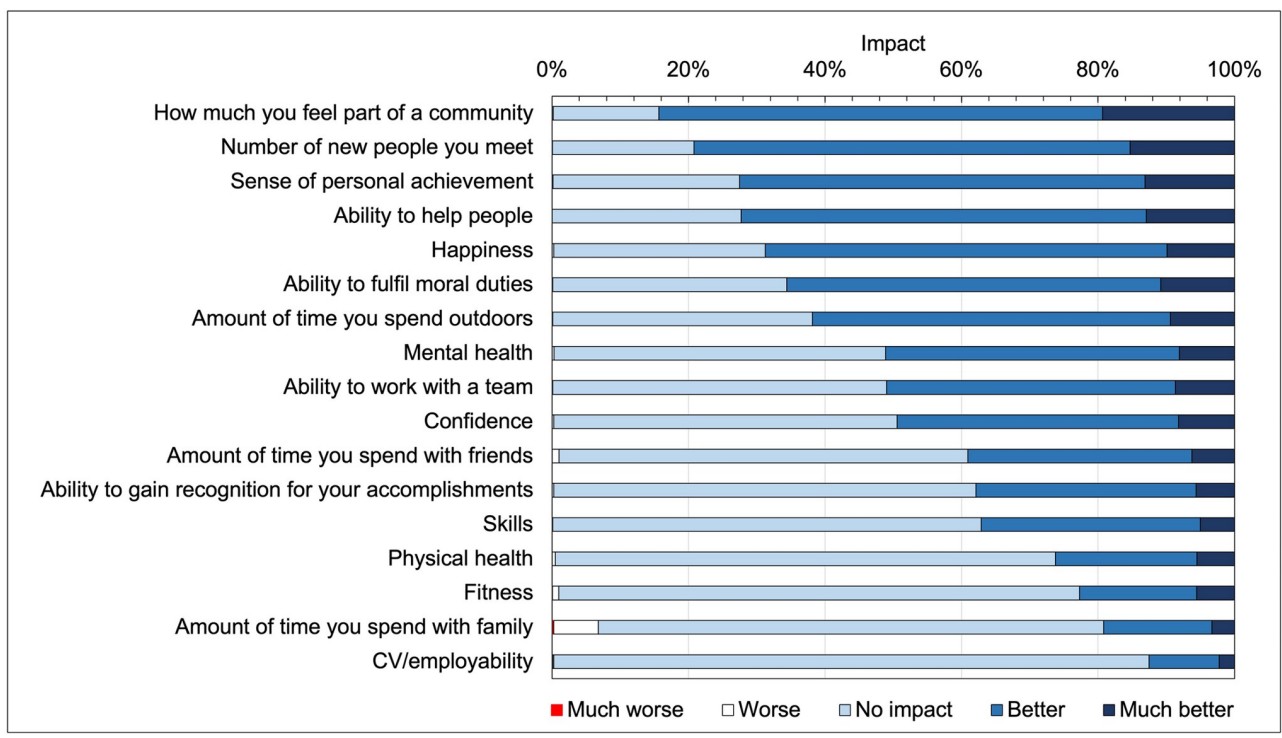

**Fig 2. Perceived impact of volunteering at *parkrun* using the question "*Thinking about the impact of parkrun on your health and wellbeing, to what extent has volunteering at parkrun changed [your/the]*".**

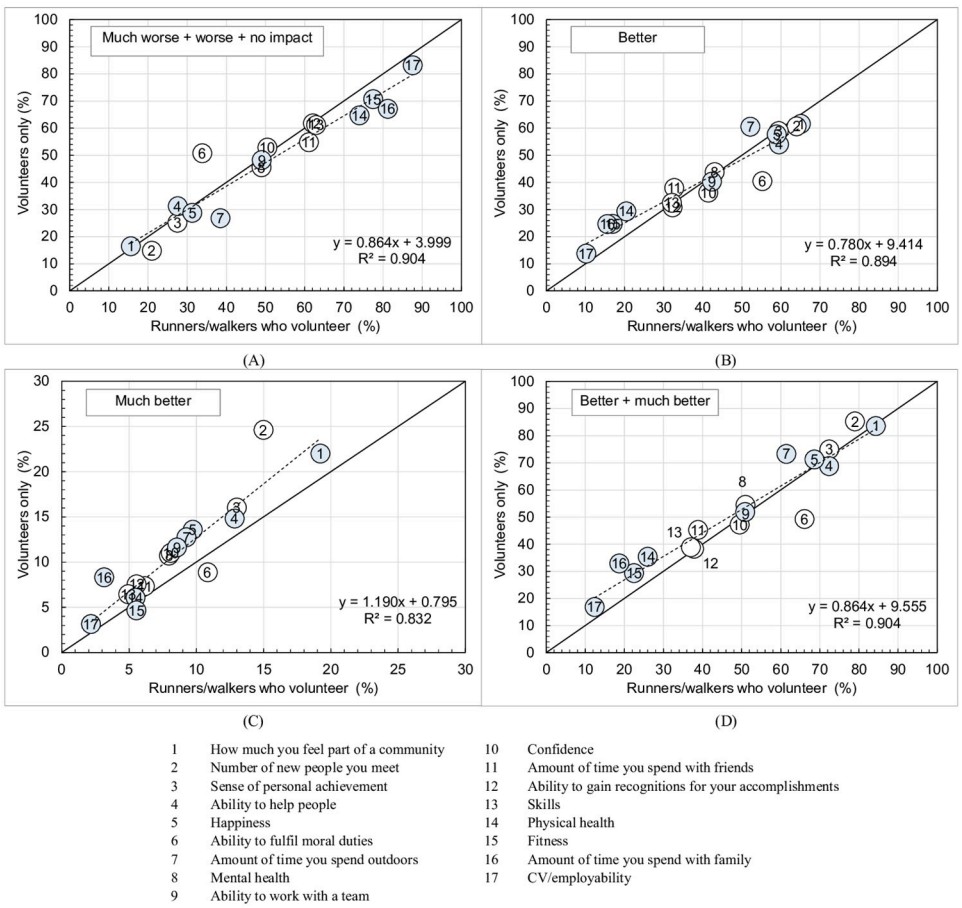

**Fig 3. Comparison of perceived impact at *parkrun* for those who exclusively volunteer compared to runners/walkers who volunteer: (A) *worse + much worse + no impact*; (B) *better*; (C) *much better*; and (D) *better + much better*.** Shading indicates distributions are different at $p<0.001$. All regressions significant at $p<0.001$. See legend for specific impacts.

26.0%, small effect size = 0.04), *fitness* (29.3 v 22.5%, small effect size = 0.04) and *amount of time you spend with family* (32.9 v 18.7%, small effect size = 0.07).

Table 4 shows that 8 out of 17 measures showed no significant difference (at $p<0.001$) between volunteers and RWV. These included *how much you feel part of a community* (84.3%), *sense of personal achievement* (72.5%), *ability to help people* (72.3%), *happiness* (68.7%) and *mental health* (51.1%).

Table 5 and Fig 4 compare impact for those who exclusively volunteer with RWDV. Some differences are evident:

1. Volunteers were more likely (due to volunteering) than RWDV (due to running/walking) to report *better + much better* for *how much you feel part of a community* (83.5 v 61.2%; $p<0.001$, small effect size = 0.05) and *the number of new people you meet* (85.2 v 44.7%; $p<0.001$, small effect size = 0.09);

2. Volunteers were less likely (due to volunteering) than RWDV (due to running/walking) to report *better + much better* for *fitness* (29.3 v 88.1%; $p<0.001$, small to moderate effect size = 0.16) and *physical health* (35.4 v 83.4%; $p<0.001$, small to moderate effect size = 0.12).

**Table 5. Perceived impact of *parkrun* from volunteering for volunteers and running/walking for runners/walkers who don't volunteer (RWDV) using the question "*Thinking about the impact of parkrun on your health and wellbeing, to what extent has [volunteering / running/walking] at parkrun changed [your/the]*".**

| | Measure | n | Worse + much worse | Much worse | Worse | No impact | Better | Much better | Better + Much Better | χ² | p | φ_c |
|---|---|---|---|---|---|---|---|---|---|---|---|---|
| | | | **Volunteers** | | | | | | | **χ² tests** | | |
| 3 | Sense of personal achievement | 630 | 25.1% | 0.0% | 0.2% | 24.9% | 58.9% | 16.0% | 74.9% | | | |
| 15 | Fitness | 624 | 70.7% | 0.2% | 1.0% | 69.6% | 24.7% | 4.6% | 29.3% | | | |
| 14 | Physical health | 628 | 64.6% | 0.0% | 1.1% | 63.5% | 29.3% | 6.1% | 35.4% | | | |
| 5 | Happiness | 626 | 28.8% | 0.2% | 0.8% | 27.8% | 57.7% | 13.6% | 71.2% | | | |
| 7 | Amount of time you spend outdoors | 628 | 26.8% | 0.0% | 0.3% | 26.4% | 60.5% | 12.7% | 73.2% | | | |
| 8 | Mental health | 626 | 45.5% | 0.3% | 1.3% | 43.9% | 43.8% | 10.7% | 54.5% | | | |
| 1 | Feel part of a community | 632 | 16.5% | 0.2% | 0.3% | 16.0% | 61.6% | 22.0% | 83.5% | | | |
| 10 | Confidence | 627 | 52.8% | 0.2% | 0.6% | 52.0% | 36.2% | 11.0% | 47.2% | | | |
| 2 | Number of new people you meet | 630 | 14.8% | 0.2% | 0.3% | 14.3% | 60.6% | 24.6% | 85.2% | | | |
| 11 | Amount of time you spend with friends | 628 | 54.8% | 0.0% | 0.6% | 54.1% | 37.9% | 7.3% | 45.2% | | | |
| 16 | Amount of time you spend with family | 626 | 67.1% | 0.0% | 5.1% | 62.0% | 24.6% | 8.3% | 32.9% | | | |
| | | | **Runners/walkers who don't volunteer** | | | | | | | | | |
| 3 | Sense of personal achievement | 35,697 | 10.0% | 0.1% | 0.4% | 9.5% | 60.0% | 30.0% | 90.0% | | | |
| 15 | Fitness | 35,694 | 11.9% | 0.1% | 0.3% | 11.6% | 65.4% | 22.7% | 88.1% | | | |
| 14 | Physical health | 35,682 | 16.6% | 0.1% | 0.5% | 16.0% | 65.4% | 18.0% | 83.4% | | | |
| 5 | Happiness | 35,653 | 24.4% | 0.1% | 0.2% | 24.1% | 63.3% | 12.3% | 75.6% | | | |
| 7 | Amount of time you spend outdoors | 35,676 | 28.4% | 0.0% | 0.1% | 28.3% | 58.0% | 13.6% | 71.6% | | | |
| 8 | Mental health | 35,651 | 33.1% | 0.1% | 0.2% | 32.8% | 54.7% | 12.2% | 66.9% | | | |
| 1 | Feel part of a community | 35,660 | 38.8% | 0.1% | 0.4% | 38.3% | 51.3% | 9.9% | 61.2% | | | |
| 10 | Confidence | 35,658 | 40.1% | 0.1% | 0.6% | 39.5% | 49.2% | 10.6% | 59.9% | | | |
| 2 | Number of new people you meet | 35,657 | 55.3% | 0.1% | 0.3% | 54.9% | 38.2% | 6.5% | 44.7% | | | |
| 11 | Amount of time you spend with friends | 35,627 | 67.0% | 0.1% | 0.9% | 66.0% | 28.3% | 4.7% | 33.0% | | | |
| 16 | Amount of time you spend with family | 35,605 | 73.9% | 0.1% | 5.4% | 68.4% | 21.7% | 4.3% | 26.1% | | | |
| | | | **All** | | | | | | | | | |
| 3 | Sense of personal achievement | 36,327 | 10.3% | 0.1% | 0.4% | 9.8% | 60.0% | 29.7% | 89.7% | 178 | <0.001 | 0.05 |
| 15 | Fitness | 36,318 | 12.9% | 0.1% | 0.3% | 12.6% | 64.7% | 22.4% | 87.1% | 1,891 | <0.001 | 0.16 |
| 14 | Physical health | 36,310 | 17.5% | 0.1% | 0.6% | 16.9% | 64.8% | 17.7% | 82.5% | 987 | <0.001 | 0.12 |
| 5 | Happiness | 36,279 | 24.5% | 0.1% | 0.2% | 24.2% | 63.2% | 12.3% | 75.5% | 8.7 | 0.069 | 0.01 |
| 7 | Amount of time you spend outdoors | 36,304 | 28.4% | 0.0% | 0.1% | 28.3% | 58.0% | 13.6% | 71.6% | 1.6 | 0.804 | 0.00 |
| 8 | Mental health | 36,277 | 33.3% | 0.1% | 0.3% | 33.0% | 54.5% | 12.2% | 66.7% | 70 | <0.001 | 0.03 |
| 1 | Feel part of a community | 36,292 | 38.4% | 0.1% | 0.4% | 37.9% | 51.5% | 10.1% | 61.6% | 183 | <0.001 | 0.05 |
| 10 | Confidence | 36,285 | 40.3% | 0.1% | 0.6% | 39.7% | 49.0% | 10.6% | 59.7% | 46.0 | <0.001 | 0.03 |
| 2 | Number of new people you meet | 36,287 | 54.6% | 0.1% | 0.3% | 54.2% | 38.6% | 6.9% | 45.4% | 562 | <0.001 | 0.09 |
| 11 | Amount of time you spend with friends | 36,255 | 66.8% | 0.1% | 0.9% | 65.8% | 28.5% | 4.7% | 33.2% | 43.2 | <0.001 | 0.02 |
| 16 | Amount of time you spend with family | 36,231 | 73.8% | 0.1% | 5.4% | 68.3% | 21.8% | 4.4% | 26.2% | 29.3 | <0.001 | 0.01 |

Measures are ranked largest to smallest by *better + much better* for the combined sample and χ² tests used a combined category for *no impact*, *worse* and *much worse* due to counts less than five in the latter two categories. Values shown for volunteers, runners/walkers who don't volunteer and all respondents.

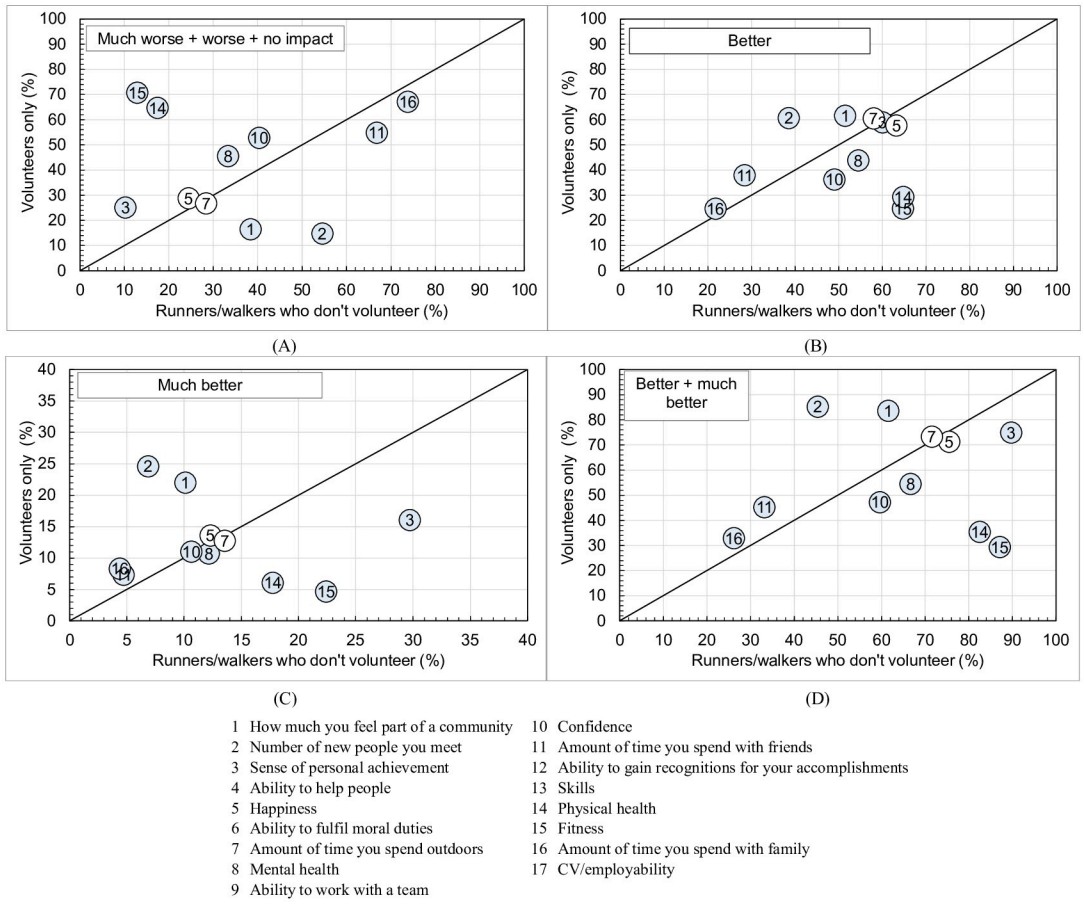

**Fig 4. Comparison of perceived impact of *parkrun* for those who exclusively volunteer compared to runners/walkers who don't volunteer: (A)** *worse + much worse + no impact*; **(B)** *better*; **(C)** *much better*; **and (D)** *better + much better*. Shading indicates distributions are different at *p*<0.001. See legend for specific impacts.

Only two measures had no significant differences between volunteers and RWDV, *happiness* (75.5%) and *amount of time you spend outdoors* (71.6%).

## Discussion

The aim of this study was to compare the impact of volunteering on people who volunteer exclusively at *parkrun* with those who run/walk and volunteer or run/walk and don't volunteer. Those who identified as volunteers were older than those participating as runners/walkers, were likely to be from a white ethnic background and were more likely to be retired. Volunteers were more likely be from less deprived neighbourhoods, which supports the findings in existing volunteering literature [1]. That said, only 30% of those who exclusively volunteer at *parkrun* came from the two lowest IMD quartiles, and organisations such as *parkrun* will still need to focus on engaging more people from deprived neighbourhoods to maximise potential gains.

Understanding who engages in volunteering and why they do so is important in the promotion of volunteering. Our findings demonstrate that people's intentions for volunteering were often realised as positive perceived impacts after volunteering. In addition to the social

benefits, people who volunteered benefitted in a variety of ways (personal achievement, happiness, amount of time spent outdoors, mental health), though these outcomes might not have been originally anticipated as self-oriented motives. Some benefits from volunteering were at least as great as from running or walking, such as happiness and the amount of time spent outdoors. While fitness and physical health improved much less for volunteers than those who ran or walked, around a third still improved.

Helping people was a key motivation for volunteering, selected by around a quarter of volunteers and RWV and their ability to do this was enabled by volunteering at *parkrun* for about 7 in 10. RWV were highly motivated by obligation and moral duty with this satisfied by around two-thirds of RWV compared to half the volunteers. It is possible that RWV balance the cost of volunteering (a median of 2.4 times per year) with the reward of runs (a median of 11.1 times per year). Volunteering for RWV also provided an opportunity to still take part during injury, illness or rest days, which has been found in previous *parkrun* research [13].

There was not always correspondence between motive and perceived impact. For instance, the number of new people met was improved by volunteering for 85.2% of volunteers, 79.0% of RWV and 44.7% of RWDV while it was ranked only moderately as a motive. Spending time with friends was ranked low as a motive for all groups, but had a moderately positive impact (45.2, 38.9 and 33.0% for volunteers, RWV and RWDV respectively).

Some motives were highly ranked and also had high impact. For instance, high proportions of both volunteers and RWV were motivated by wanting to feel part of a community and over 8 out of 10 volunteers, regardless of whether the also ran or walked reported this as better or much better. Conversely, RWDV were less motived by this and fewer reported positive benefits.

Many of the volunteering roles at *parkrun* involve interacting with people and volunteers volunteered almost five times as often as those who also ran or walked. This suggests that social connections are more frequently enhanced through volunteering at *parkrun*, which has the potential to bridge and link people across social boundaries and socioeconomic divides [20, 24].

Improving fitness through volunteering was selected by a small proportion of volunteers and RWV as a motive, but 29.3% of volunteers and 22.5% of RWV reported it had improved due to volunteering. Volunteering tasks may require volunteers to walk to positions that could be up to 2.5 km away (for an out-and-back *parkrun* course) while most will be standing for the time of the slowest parkrunner (sometimes more than an hour). More volunteers than RWV were motivated by spending time outdoors (17.2 v 4.4%) and more volunteers than RWV perceived a benefit from being outdoors when volunteering (73.2 v 61.5%). This may be because RWV already spend time outdoors through their running or walking.

Learning new skills and enhancing one's CV was low on motives for volunteering at *parkrun*. In the UK, younger people volunteer as part of the Duke of Edinburgh Award Scheme as a way of learning new skills; however, these, would have been excluded by the age limit for the survey of 16 years. Despite this, around 37–39% reported improvement to their skills.

## Wellbeing outcomes

The data shows that volunteers at *parkrun* experience four of the components of the Five Ways to Wellbeing model: (1) increased physical activity; (2) connections with others; (3) giving; and (4) learning new skills. The remaining component–mindfulness–was not explored in the study but would be worthy of further investigation. While mental and physical health were ranked low as motives for volunteering, over half of the volunteers reported improvements to their mental health and around a quarter reported improvements to their physical health. A

larger proportion of volunteers than RWV improved their physical health from volunteering. Additionally, a larger proportion of volunteers reported improvement to feeling part of a community or meeting new people than RWDV. These findings suggest that feeling part of community can be enhanced if people volunteer at *parkrun* rather than see it only as a running/walking event.

Our findings support previous *parkrun* volunteering research [13] which showed the primary motive for volunteering was to 'give something back'. Wiltshire and Stevinson [25] explored this through the lens of social capital (the resources and links that derive from contacts in society). The authors talked about *parkrun* volunteering as a way for participants to benefit from the aggregate labour of the wider community. They described how the format of *parkrun* events means that those taking part as runners/walkers feel not only gratitude to the volunteers (and the organisation) but feel a sense of debt to them.

Our findings support the notion that by increasing social and community connectedness, *parkrun* fosters a desire to give back to the organisation (e.g., through volunteering), thereby demonstrating a capacity to mobilise resources, and promote *parkrun* participation, through social networks [25]. Understanding the wider impacts of this reciprocal volunteering behaviour at *parkrun* on community social capital could be particularly relevant in rebuilding individuals and communities in the COVID-19 era and beyond. Further research should seek to demonstrate the impact and economic value of *parkrun* volunteering, especially in the years following the COVID-19 pandemic. Attention should be given to the monitoring of outcomes and measuring the social impact using social return on investment (SROI) [26].

## Implications for practice

The example of *parkrun* shows that volunteering can impact positively on the sense of connectedness with the community. Given the positive implications of community connectedness for individual health promotion and potential wider benefits for the community and social capital [16], it is not surprising that *parkrun* has been promoted as a social prescribing offer by the Royal College of General Practitioners [27]. Launched in the UK in 2018, the '*parkrun* practice' initiative involves General Practices linking with their local *parkrun* event, with practice staff promoting *parkrun* to their patients, patient carers and colleagues [24, 28]. *parkrun* offers social prescribers a 'service', or community asset, that is local, accessible, regular, permanent, optional, and welcoming of people of all ages, backgrounds and abilities [15]. These characteristics are transferable to other community events so that community activity providers and volunteer organisations can create events that support social prescribing by healthcare practitioners.

Further research is needed to understand the appeal of participation from the perspective of patients and public health staff to understand the barriers faced by people from more diverse groups. Research needs to explore whether *parkrun* can provide a social prescription option that is appropriate for diverse populations. Given the various ways in which people can participate and different patient groups for which the signposting could be made [29, 30], such research could perhaps take a realist approach to explore (1) what works, (2) for whom it works, (3) under what circumstances it works and (4) how it works.

## Methodological considerations

This study has strengths, including the large sample size allowing for comparison between different types of *parkrun* participation. The downside of such large numbers are that statistical differences are possible where effects are relatively small. We have attempted to be realistic about effect sizes (e.g. <0.1 is small) but, being mindful that even low values of effect size when

significant can be impactful to population health if numbers are large enough, we have suggested that and effect size >0.1 is small to moderate in the context of parkrun. This assessment is a subjective one and open to debate.

Overall, the findings should be interpreted in the context of the following methodological considerations. The self-selected sample may have attracted those who had experienced higher levels of perceived benefits of *parkrun* and the findings should be interpreted with this bias in mind. The data used in this analysis are based on self-report data bearing the risk of self-report bias.

We were not able to differentiate those who volunteered at *parkrun* on an episodic basis and those who had more sustained long-term volunteer roles (e.g., Ambassadors who help set up events or support outreach programmes), which would be worthwhile for future research to explore in the context of understanding more about volunteer management and ensuring the sustainability of the events. The survey only asked participants about their motives for first volunteering at *parkrun* and, as research suggests that initial motives might not be enough to sustain participation in the long-term [7], it would be worthwhile to monitor *parkrun* volunteer motives and incentives for continued participation over time. The survey was only available in online format in the English language which may have excluded people who had limited internet access or low literacy and digital literacy levels.

## Conclusions

Large proportions of *parkrun* participants identifying as exclusively volunteers reported improvements to different aspects of their health and wellbeing. Volunteers were much less likely than runners/walkers who also volunteer to be motivated by a feeling of obligation or moral duty, but equally likely to volunteer to help people or feel part of a community. Volunteers were more likely to report improvements from volunteering than runners/walkers who volunteer for impacts relating to connections with others; examples were feeling part of a community, meeting new people and spending time with family. While improving mental and physical health was ranked low as a motive for volunteers, over half reported improvements due to volunteering at *parkrun* to mental health, and a quarter to physical health. The data shows that volunteering at *parkrun* without participating as a runner or walker can deliver some of the components of the *Five Ways to Wellbeing* advocated by the NHS. The characteristics of *parkrun* (free, regular, local, accessible and optional) make it a viable social prescribing offer and can be used as a model for other community events seeking to attract volunteers and do the same.

## Supporting information

**S1 Fig.**
(JPEG)

**S1 File. Health and wellbeing survey.**
(PDF)

**S2 File. Analysis of walking and running instances in volunteers.**
(DOCX)

## Acknowledgments

The authors would like to thank all the *parkrun* participants who completed the survey and to thank the *parkrun* Research Board for their support and guidance. Thanks also go to Chrissie

Wellington from *parkrun* for supporting the project and Mike Graney for providing parkrun participation data to allow the research to be carried out. The authors would also like to thank Allison Dunne (Sheffield Hallam University) and Rachel Hallett (The Open University) for their helpful feedback and critique of the manuscript.

## Author Contributions

**Conceptualization:** Steve Haake, Helen Quirk.

**Data curation:** Steve Haake, Alice Bullas.

**Formal analysis:** Steve Haake, Alice Bullas.

**Funding acquisition:** Steve Haake.

**Investigation:** Steve Haake.

**Methodology:** Steve Haake, Helen Quirk, Alice Bullas.

**Project administration:** Helen Quirk, Alice Bullas.

**Resources:** Helen Quirk.

**Supervision:** Steve Haake.

**Validation:** Alice Bullas.

**Visualization:** Steve Haake.

**Writing – original draft:** Steve Haake, Helen Quirk.

**Writing – review & editing:** Steve Haake, Helen Quirk.

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
