## [Decision Letter · Decision Letter 0]

23 Nov 2021

PGPH-D-21-00850

The health benefits of volunteering at a free, weekly, 5 km event in the UK: a cross-sectional study of volunteers at parkrun

Dear Dr. Haake,

Thank you for submitting your manuscript to PLOS Global Public Health. After careful consideration, we feel that it has merit and some minor changes are needed to fully meet PLOS Global Public Health’s publication criteria. Therefore, we invite you to submit a revised version of the manuscript that addresses the points raised during the review process.

We look forward to receiving your revised manuscript.

Kind regards,

Chunxiao Li, Ph.D.

Academic Editor

Journal Requirements:

1. Peer review at PLOS ONE is not double-blinded (https://journals.plos.org/plosone/s/editorial-and-peer-review-process). For this reason, authors should include in the revised manuscript all the information removed for blind review.

2. Please update the completed 'Competing Interests' statement, including any COIs declared by your co-authors. If you have no competing interests to declare, please state "The authors have declared that no competing interests exist". Otherwise please declare all competing interests beginning with the statement "I have read the journal's policy and the authors of this manuscript have the following competing interests:"

3. We have noticed that you have uploaded supporting information but you have not included a list of legends.  Please add a full list of legends for all supporting information files (including figures, table and data files) after the references list. 

4. In the online submission form, you indicated that "Datasets relating to this manuscript are stored in the Sheffield Hallam University Research Database (SHURDA). Access is allowed in accordance with the Data Protection Act 2018 and the General Data Protection Regulation 2018. A copy of the full sample of anonymised data used in the manuscript is a available to researchers through the parkrun Research Board as outlined in the participant information sheet, subject to ethics approval and a data sharing agreement. Contact the corresponding author for access details."

5. Please ensure you include your funder's role statement "The funders had no role in study design, data collection and analysis, decision to publish, or preparation of the manuscript." at the end of your amended statement

Reviewers' comments:

Reviewer's Responses to Questions

**Comments to the Author**

1. Does this manuscript meet PLOS Global Public Health’s publication criteria? Is the manuscript technically sound, and do the data support the conclusions? The manuscript must describe methodologically and ethically rigorous research with conclusions that are appropriately drawn based on the data presented.

Reviewer #1: Yes

Reviewer #2: Yes

2. Has the statistical analysis been performed appropriately and rigorously?

Reviewer #1: Yes

Reviewer #2: Yes

3. Have the authors made all data underlying the findings in their manuscript fully available (please refer to the Data Availability Statement at the start of the manuscript PDF file)?

Reviewer #1: Yes

Reviewer #2: Yes

4. Is the manuscript presented in an intelligible fashion and written in standard English?

Reviewer #1: Yes

Reviewer #2: Yes

5. Review Comments to the Author

Reviewer #1: Very well written submission. The manuscript contained a very detailed methodology and thorough explanation of the implementation which was appropriate to fully meet the overall objectives of the study. The large data set seems to have been rigorously analysed and support the conclusions. The majority of the results were presented in a clear and easily understandable format. However, figure 2 "Perceived impact of volunteering at parkrun" seems to be missing which should be amended.

Reviewer #2: The manuscript is nicely written by the authors. It is very interesting work on the volunteers.

The methods used by the authors are appropriate and can be replicated easily. The analysis and interpretation is ok and discussed in multiple angles.

6. PLOS authors have the option to publish the peer review history of their article (what does this mean?). If published, this will include your full peer review and any attached files.

**Do you want your identity to be public for this peer review?** For information about this choice, including consent withdrawal, please see our Privacy Policy.

Reviewer #1: No

Reviewer #2: **Yes: **Palash Chandra Banik

---

## [Editor Report · Decision Letter 1]

14 Dec 2021

The health benefits of volunteering at a free, weekly, 5 km event in the UK: a cross-sectional study of volunteers at parkrun

PGPH-D-21-00850R1

Dear Dr. Haake,

We're pleased to inform you that your manuscript has been judged scientifically suitable for publication and will be formally accepted for publication once it meets all outstanding technical requirements.

Within one week, you'll receive an e-mail detailing the required amendments. When these have been addressed, you'll receive a formal acceptance letter and your manuscript will be scheduled for publication.

An invoice for payment will follow shortly after the formal acceptance. To ensure an efficient process, please log into Editorial Manager at https://www.editorialmanager.com/pgph/ click the 'Update My Information' link at the top of the page, and double check that your user information is up-to-date. If you have any billing related questions, please contact our Author Billing department directly at authorbilling@plos.org.

Kind regards,

Chunxiao Li, Ph.D.

Academic Editor
